# Design and Parameters Optimization of a Provoke-Suction Type Harvester for Ground Jujube Fruit

Gaokun Shi [1,2], Jingbin Li [1,2,*], Za Kan [1,2], Longpeng Ding [1,2], Huizhe Ding [1,2], Lun Zhou [1,2] and Lihong Wang [3]

1    College of Mechanical and Electrical Engineering, Shihezi University, Shihezi 832000, China; shigk@stu.shzu.edu.cn (G.S.); kz_mac@shzu.edu.cn (Z.K.); dy2016@shzu.edu.cn (L.D.); dinghz@stu.shzu.edu.cn (H.D.); zhoulun@stu.shzu.edu.cn (L.Z.)
2    Xinjiang Production and Construction Corps, Key Laboratory of Modern Agricultural Machinery, Shihezi 832000, China
3    College of Engineering and Technology, Southwest University, Chongqing 400715, China; w20190305@swu.edu.cn
*    Correspondence: lijingbin@shzu.edu.cn

**Abstract:** Low working efficiency is an important reason for the limited application of the traditional aspirated-air type jujube fruit pickup machine. In this study, a provoke-suction type harvester for ground jujube fruit (PSH) was designed, based on the principle of negative pressure suction after shoveling the jujube fruit mechanically. The main working parameters were analyzed and the structures of the key devices were designed. Then, a three-factor and three-level Box–Behnken method was used to evaluate the performance of the PSH. The results showed that the pickup rate, impurity rate, and working efficiency were 99.36%, 5.63%, and 1672.3 kg·h$^{-1}$, respectively; as the forward speed, provoke teeth buried depth, and airflow velocity were 0.21 kg·h$^{-1}$, 74 mm, and 26.4 kg·h$^{-1}$, respectively. Furthermore, the verification results showed that the pickup rate, impurity rate, and working efficiency were 98.05%, 5.97%, and 1591.2 kg·h$^{-1}$, respectively, moreover, the relative errors were 1.32%, 6.04%, and 4.85%, respectively, indicating that the parameter optimization model can accurately predict the test results. The working efficiency of the PSH was significantly improved compared with the traditional aspirated-air type jujube fruit pickup machine. This research can provide a reference for the development of the jujube fruit pickup machine.

**Keywords:** agricultural machinery; jujube fruit; pick up; parameters optimization

## 1. Introduction

The Xinjiang Province is the main production area for high-quality jujube fruit (*Ziziphus jujube* Mill.) in China, accounting for about 3.81 million tons (50%) of the national total jujube fruit production in 2020 [1]. The main cultivated varieties are Jun jujube fruit and Grey jujube fruit, which are mainly utilized to process into dried jujube fruit or deep-processing production [2]. Hence, they are usually dried naturally on the tree to increase the nutritional components and improve the quality of the jujube fruit [3,4]. At this time, the connecting force between fruit stalk and jujube fruit is small. Hence, many jujube fruit will fall off with the disturbance of external environmental factors, such as wind and rain [5]. Therefore, the existing jujube fruit harvesting process is to: (1) knock off the jujube fruit which remain on the tree, (2) collect the jujube fruit to the row's middle to form a "jujube fruit belt", and (3) pick up jujube fruit manually. Traditional harvesting of fruit is an extremely laborious, time-consuming, and labor-intensive operation. Modern agriculture is shifting from tedious manual harvesting to a continuously automated operation [6,7].

Many researchers are focused on developing various machinery for picking up jujube fruit. They mainly include mechanical type [8–12] and aspirated-air type jujube fruit pickup machines [13–17], according to the operating principles. The working principles of the mechanical type jujube fruit pick up machine were using a shovel, pick, pull, clip, and other

operation forms to pick up jujube fruit [18–20]. Zhang et al. [9] designed a pickup device driven by an eccentric wheel to shovel the jujube fruit on the ground. Li et al. [12] studied a roller-type jujube fruit pickup machine, which transfers the ground jujube fruit to the conveyor belt with a pickup drum which installed a flexible paddle, and removed the impurities with airflow and poke rod. Lu et al. [11] employed a cleaning roller brush arranged to gather jujube fruit that formed the "jujube fruit belt". Then a shovel device is used to collect jujube fruit, and the impurities such as soil blocks and stones leak out from the spacing gap of the shovel device. The results showed that the working efficiency is more than 400 kg·h$^{-1}$ and the pickup rate is more than 94%. The mechanical type jujube fruit pickup machines have a high working efficiency, but they easily caused the jujube fruit damage during the picking up process, and it is difficult to effectively remove the impurities contained in jujube fruit.

The working principle of the aspirated-air type jujube fruit pickup machine was to pick up jujube fruit by using the negative pressure airflow. Shi et al. [16] use venturi to convert the positive pressure airflow generated by the fan into negative pressure airflow to suck up jujube fruit, based on Bernoulli's principle. Then, the cleaning box is used to remove the impurities by utilizing the specific gravity difference between jujube and impurities. The results showed that the pickup rate is 182.8 kg·h$^{-1}$. Zhang et al. [13] developed a pneumatic pickup machine for low-density cultivation of jujube orchard by using the negative pressure airflow generated by the centrifugal fan to pick up jujube fruit and remove impurities through a vibrating screen. The results showed that the pickup rate and impurity rate were 96.41% and 1.54%, respectively. Zhang et al. [15] conducted an air suction type picker for ground jujube fruit by the same method as Zhang et al. [13]. The results showed that the working efficiency, impurity rate, and pickup rate were 220 kg·h$^{-1}$, 3.75%, and 92.20%, respectively, in the Jun jujube orchard, and 285 kg·h$^{-1}$, 4.28%, and 90.65% in the Grey jujube orchard. The aspirated-air type jujube fruit pickup machine can better pick up jujube fruit and remove impurities, but the suction inlet needs to be kept at a certain distance to the ground manually, and there are the disadvantages of fast airflow dissipation and low airflow utilization rate. Hence, the working efficiency usually is 100–500 kg·h$^{-1}$ [16–20].

Therefore, in this paper, combined with the advantages of the mechanical type and aspirated-air type jujube fruit pickup machine, a provoke-suction type harvester (PSH) for ground jujube fruit was designed which is based on the principle of mechanical shoveling and negative pressure airflow suction jujube fruit. The Box–Behnken method was used to analyze the performance of the PSH. This paper can provide a new harvesting method and technical reference for the research of jujube fruit pickup machinery and/or equipment.

## 2. Materials and Methods

### 2.1. Structure of the PSH

Figure 1 shows the structure of the PSH, which mainly consists of the control system, cleaning device, diesel engine, centrifugal fan, pickup device, and caterpillar chassis. The pickup device was connected to the inlet of the cleaning device through the suction pipe, and the airflow outlet of the cleaning device was connected with the suction inlet of the centrifugal fan. The control system was utilized to control the operation of the working device and drive the crawler chassis. The centrifugal fan and hydraulic pump were driven by the diesel engine through the vee belt. The hydraulic pump provides a power source for the working device and the caterpillar chassis.

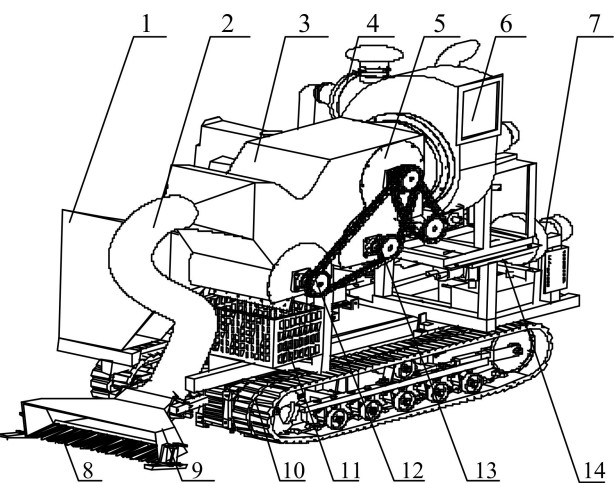

**Figure 1.** Structure of the provoke-suction type harvester for ground jujube fruit (PSH). 1. Control system 2. Suction pipe 3. Cleaning device 4. Diesel engine 5. Rotary screen 6. Centrifugal fan 7. Oil hydraulic pump 8. Provoke teeth 9. Pickup device 10. Caterpillar chassis 11. Basket 12. Closed-air aspirator of discharge jujube fruit 13. Closed-air aspirator of discharge impurities materials 14. Rack.

*2.2. Working Principle*

During operation, with the forward movement of the PSH, the provoking teeth tip of the pickup device will penetrate the soil to shovel the jujube fruit and gradually gather on the pickup device. The negative pressure airflow was transmitted to the pickup device through the cleaning device and suction pipe when the centrifugal fan was running. Jujube fruit will move and enter the cleaning device along with the suction pipe at the action of negative pressure airflow. Then, the jujube fruit and impurities will form a different motion trajectory, due to the specific gravity and fluid characteristics difference and the action of variable cross-section structure of the cleaning device. The jujube fruit were discharged from the closed-air aspirator for discharge jujube fruit and fell into the basket, as they settled in front of the cleaning device. The impurities continue to move and are discharged through the closed-air aspirator for discharge impurities materials, as blocked by the rotary screen. Thus, the process of pickup jujube fruit and cleaning impurities was completed by the PSH.

*2.3. Operating Conditions and Main Technical Parameters*

Before harvesting, the jujube fruit were manually collected to the middle of jujube tree rows and formed the "jujube fruit belt" of which width was less than 1.0 m. The operation condition of the PSH was to pick up jujube fruit from the "jujube fruit belt" and remove the impurities (mainly jujube leaves and bearing branches). Combined with the requirements of the jujube fruit harvesting operation, the main technical parameters of the PSH were determined (Table 1).

**Table 1.** Main technical parameters of the provoke-suction type harvester for ground jujube fruit (PSH).

| Items | Values/Type |
|---|---|
| Rated horsepower/kW | 36.8 |
| Unity machine dimensions (Length × width × height)/(mm × mm × mm) | 2840 × 1320 × 2130 |
| Working width/m | 1.0 |
| Drive type | Hydraulic drive |
| Centrifugal fan model | Y5-47 |
| Forward speed/(km·h$^{-1}$) | 0–1.5 |

### 2.4. Parameters Analysis and Device Design

2.4.1. Critical Velocity

Only when the airflow velocity is greater than the critical velocity of jujube fruit, can they be sucked up and transported. Therefore, the critical velocity is the minimum airflow velocity to ensure the normal operation of the PSH [21,22].

The established force balance equation of jujube fruit weight and airflow force is:

$$F_a = m_p g \tag{1}$$

where $F_a$ is the airflow force for jujube fruit, N; $m_p$ is the weight of jujube fruit, kg; g is the acceleration of gravity, m·s$^{-2}$.

The expression of drag force $F_a$ is:

$$F_a = \frac{1}{2} C A \rho_f u_g{}^2 \tag{2}$$

where $C$ is the drag coefficient; $A$ is the projected area of jujube fruit, m$^2$; $\rho_f$ is the air density, and the value is 1.205 kg·m$^{-3}$ at 20 °C; $u_g$ is the airflow velocity, m·s$^{-1}$.

The value of $C$ was determined by the Reynolds number, and the calculation equations are:

$$C = \begin{cases} \frac{24}{Re}, & 1 \leq Re_p \\ \frac{24(1+0.25Re^{0.687})}{Re}, & 1 < Re_p < 10^3 \\ 0.44, & Re_p > 10^3 \end{cases} \tag{3}$$

$$Re = \frac{\varepsilon d_v \rho_f u_g}{\eta}$$

$$d_v = 1.24 \left( \frac{m_n}{\rho_s n} \right)^{\frac{1}{3}}$$

where Re is the Reynolds number after introducing porosity; $d_v$ is the equivalent diameter of jujube fruit, m; $\eta$ is air viscosity coefficient, m$^2$·s$^{-1}$, and the value is $1.5 \times 10^{-5}$ m$^2$·s$^{-1}$ at 20 °C; $\varepsilon$ is the porosity, %; $m_n$ is the mass sum of N jujube fruit, kg; $\rho_f$ is the density of jujube fruit, kg·m$^{-3}$.

By substituting the jujube fruit parameters into Equation (3), it can be known that Re is greater than $10^3$. So, the value of $C$ is 0.44. Then, substituting 0.44 into Equation (2) can be obtained Equation (4):

$$u_{(g.min)} = \sqrt{\frac{m_p g}{0.22 A \rho_f}} \tag{4}$$

To ensure all jujube fruit can be picked up, it should be selected as the jujube fruit with the smallest windward area and the largest mass. According to the previous measurement of the jujube fruit physical parameters, the maximum mass ($8.21 \times 10^{-3}$ kg) and the minimum projected area were ($4.2 \times 10^{-4}$ m$^2$) substituted into Equation (4). The critical velocity was obtained to be 24.88 m·s$^{-1}$.

2.4.2. Design of the Pickup Device

(1)  Pickup Device Structure

Figure 2 shows the pickup device structure, which mainly consists of provoke teeth, underside baffle, lower baffle, negative pressure airflow interface, angle adjusting rod, side baffle, depth limiting sliding plate, etc. The pickup device was hinged on the rack and controlled by a hydraulic cylinder [23]. The angle adjusting rod was used to adjust the angle between the pickup device and the ground, to meet the need of different soil types and the flatness of the jujube orchard. The depth limit slide plate was used to adjust the depth of provoking teeth into the soil and has the function of profile.

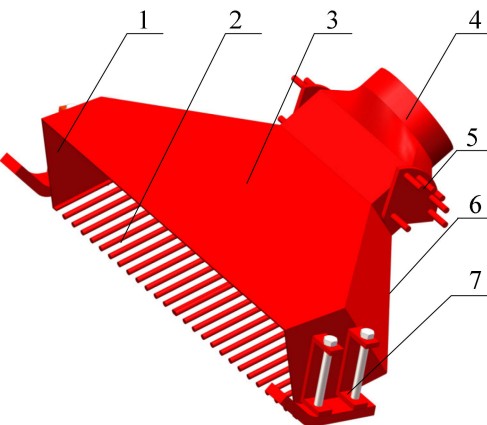

**Figure 2.** Structure diagram of the pickup device. 1. Side baffle 2. Provoking teeth 3. Upper baffle 4. Negative pressure airflow interface 5. Angle adjusting rod 6. Underside baffle 7. Depth limiting sliding plate.

(2)    Depth of the Provoking Teeth into Soil

The jujube orchard has been watered and fertilized many times, resulting in potholes and unevenness on the ground. During operation, the depth of the provoking teeth into the soil should be as shallow as possible, so as to reduce shoveling more soil, meanwhile, ensuring that the provoking teeth can pick up the jujube fruit in the lowest depression. Hence, the ground flatness of the jujube orchard was the crucial parameter to determine the provoke teeth buried depth. A rectangular area with a length of 600 mm was randomly selected at the rows of jujube fruit trees (row spacing: 3000 mm) and divided into many (50 × 50) mm square, in the jujube orchard of the whole process mechanization demonstration base of jujube fruit in the 13th regiment of Alar city, Xinjiang Province, China. Then, the vertical distance between the center point of the squares and the horizontal plane was measured and drew the cloud diagram of ground flatness (Figure 3).

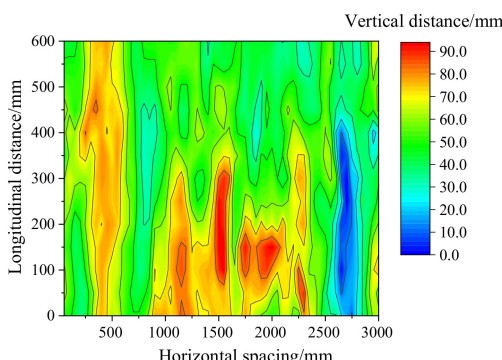

**Figure 3.** Ground flatness cloud atlas of a jujube fruit orchard.

The maximum distance between the ground plane and the lowest depression was 94 mm, and the average was 54 mm. Considering the influence of the PSH trapped in the soil and excluding the special measurement values, the maximum depth of the provoke teeth into the soil was determined to be 90 mm (it is the maximum distance between the tip of the provoke teeth and the walking plane of the PSH).

(3)    Space between the Provoke Teeth

If the spacing between the provoke teeth was too large, jujube fruit would not be shoveled. If it was too small, more impurities would be shoveled. The value of the spacing between the provoke teeth can be determined by the minor axis size of jujube fruit. A total of 500 Grey jujube fruit were randomly selected from the jujube orchard of

the whole process mechanization demonstration base of jujube fruit in group 13, Alar city, Xinjiang, China. The minor size of the jujube fruit was measured with a digital display vernier caliper (measuring range: 150 mm, measuring accuracy: 0.02 mm) according to the method of Mahawar, M. et al. [24], and the size distribution were plotted in Figure 4. The minimum size of the minor axis was 15.3 mm, so the spacing between the provoke teeth was determined to be 15 mm. The teeth tip may collide with stones, stumps, and other sundries in the soil during the operation. To ensure the provoking teeth strength, 20 Mn round steel with a diameter of 8 mm was selected [25].

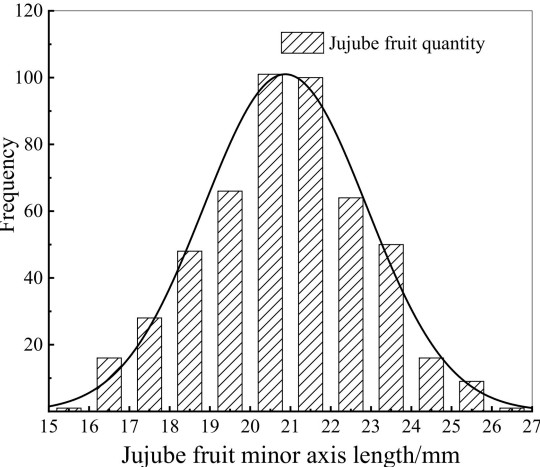

**Figure 4.** The minor axis size distribution of 500 jujube fruit.

(4)    Angle between the Provoke Teeth and Ground

The function of the angle between the picking teeth and ground was to make the jujube fruit enter the suction pipe with a small airflow velocity action. It can be seen from Equation (5) that the airflow velocity was inverse proportion to the flow section. Since the airflow velocity in the suction pipe was known from Section 2.4.1, the airflow velocity at the pickup device can be calculated by the ratio of the cross-sectional of the pickup device and suction pipe. Then, the specific value of the angle between the provoke teeth and ground can be obtained according to the force balance equation of jujube fruit on the provoke teeth [26].

$$\frac{u_p}{u_{g \cdot p}} = \frac{A_c}{A_p} \tag{5}$$

where $u_{g \cdot p}$ is the airflow velocity in the pickup device, m·s$^{-1}$; $A_c$ is the cross-sectional area of the suction pipe, mm$^2$; $A_p$ is the cross-sectional area in the pickup device, mm$^2$.

The force diagram of jujube fruit on the provoke teeth was shown in Figure 5. The jujube fruit can be picked up, only the force of airflow on the jujube fruit was greater than the sum of tangential gravity and friction of the jujube fruit along the provoke teeth direction.

Where $F_{a \cdot p}$ is the airflow force of jujube fruit in the pickup device, N; $f$ is the friction between jujube fruit and teeth picking, N; $\theta$ is the angle between the provoke teeth and the ground, (°); $G$ is the mass of a jujube fruit, kg; $G_t$ is the tangential force of the mass of a jujube fruit, kg; $G_n$ is the normal force of mass of a jujube fruit, kg; $F_n$ is the supporting force of the provoking teeth on the jujube fruit.

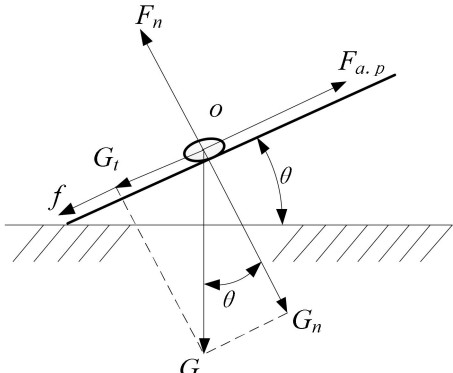

**Figure 5.** Force diagram of jujube fruit on the provoke teeth.

The mechanical equation of jujube fruit along with the provoking teeth direction was established as follows:

$$F_{a \cdot p} \geq G_t + f \tag{6}$$

The equations of $F_{a \cdot p}$, $G_t$, and $f$ are:

$$
\begin{aligned}
F_{a \cdot p} &= \tfrac{1}{2} C A_p \rho_f u_{g \cdot p}{}^2 \\
G_t &= m_p \mathrm{g} \sin \theta \\
f &= \mu m_p \mathrm{g} \cos \theta
\end{aligned}
\tag{7}
$$

where $\mu$ is the friction between jujube fruit and the provoke teeth.

By combining Equations (6) and (7), Equation (8) can be obtained:

$$\frac{1}{2} C A_p \rho_f u_{g \cdot p}{}^2 \geq m_p \mathrm{g} \sin \theta + \mu m_p \mathrm{g} \cos \theta \tag{8}$$

Substituting these values, the angle between the provoke teeth and ground was calculated to be $18°$.

### 2.4.3. Design of the Cleaning Device

Figure 6 is the structure diagram of the cleaning device, which consists of the cleaning tank, closed-air aspirator to discharge jujube fruit, closed-air aspirator to discharge impurities, drum screen, baffle, etc. The upper surface of the cleaning tank was an arc curved surface, and the box was processed with a 2 mm thick steel plate. The baffle was located between the closed-air aspirator for discharged jujube fruit and the closed-air aspirator for discharged impurities materials, its function was to change the airflow movement characteristics and further change the migration track of jujube fruit and impurities. The hydraulic motor drives the roller screen and the closed-air aspirator to rotate. The closed-air aspirator can discharge the jujube fruit and impurities continuously, and ensure the airtightness of the cleaning tank.

The velocity of jujube fruit reaches the maximum in the inlet of the cleaning device with the action of the airflow in the suction pipe [27]. Although the greater the jujube fruit velocity in the inlet of the cleaning device, the higher the working efficiency. Due to the limited structure of the cleaning device, it is difficult to settle in a short distance. Therefore, the relationship between the structural parameters of the cleaning device and the airflow velocity can be analyzed by establishing the force balance equation of the airflow action and the jujube fruit movement in the cleaning device. Since the airflow domain in the suction pipe and the cleaning box were closed, the airflow of each section was equal [28].

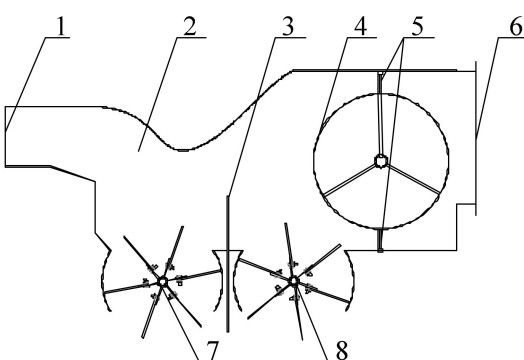

**Figure 6.** Structure diagram of the cleaning device. 1. Inlet 2. Cleaning tank 3. Baffle 4. Drum screen 5. Strip brush 6. Connecting port for centrifugal fan 7. Closed-air aspirator for discharge jujube fruit 8. Closed-air aspirator for discharge impurities.

The jujube fruit movement was a flat throwing movement with initial velocity as jujube fruit entered the cleaning device. Hence, the equation of motion of the jujube fruit is:

$$\begin{cases} l = u_{in}t \\ h = \frac{1}{2}gt^2 \end{cases} \tag{9}$$

where $l$ is the horizontal distance from the inlet to the baffle plate, m; $h$ is the vertical distance from the inlet to the baffle plate, m; $u_{in}$ is the velocity of jujube fruit entering the cleaning device, m·s$^{-1}$; $t$ is time, s.

The movement velocity of jujube fruit entering the material inlet of the cleaning box was determined by the airflow force in the suction pipe. The relationship between jujube fruit movement velocity and airflow velocity can be established:

$$\begin{cases} u_{g.max} = \sqrt{\frac{m_p(a_p+g)}{0.22A\rho_f}} \\ u_{in} = \sqrt{2s_pa_p} \end{cases} \tag{10}$$

where $u_{g.max}$ is the maximum airflow velocity, m·s$^{-1}$; $s_p$ is the length of the suction pipe that the design value is 1.6 m; $m_p$ is the mass of jujube fruit, kg; $a_p$ is the acceleration of jujube fruit, m·s$^{-2}$.

The designed value of the cleaning device length was 1.2 mm, which was determined by the size of the PSH structure. The baffle was installed between the closed-air aspirator to discharge jujube fruit and the closed-air aspirator to discharge impurities, so the horizontal distance from the inlet of the cleaning device to the baffle was 0.6 m. The distance from the feed inlet of the cleaning box to the top of the baffle was 0.15 m, which was determined by the previous research on the operation performance of the cleaning device. The maximum acceleration of jujube fruit was 7.50 m·s$^{-2}$ and the corresponding maximum airflow velocity was 35.12 m·s$^{-1}$, by substituting the values into Equations (9) and (10).

### 2.5. Test Materials

The operation performance of the PSH was carried out in the whole process mechanization demonstration base of jujube fruit in the 13th regiment of Alar city, Xinjiang Province, China on 25 November 2020 [29,30]. The variety is Xinzheng Grey jujube fruit, and the tree age was 10 years. The jujube orchard area was 220 m long and 55 m wide. The row spacing, the plant spacing, and the average plant height of jujube fruit trees were 3 m, 1.5 m, and 2.3 m, respectively. The yield of the jujube orchard was 5100 kg·ha$^{-1}$. The moisture content of the jujube fruit was 33.09% (W.B) which was measured with a Sartoriusma 100 electronic rapid moisture meter (mass accuracy: 0.001 g, accuracy: 0.01%). The ground was relatively flat, and the soil type was sandy loam.

*2.6. Test Methods*

According to the test method of DG/T 188-2019 "fruit picker" [31], several test areas with an interval of 5 m and a length of 30 m were randomly selected between the rows of jujube trees. To avoid mutual interference between the test results in the test areas, the jujube fruit and impurities in the interval areas were removed before the tests. After adjusting to the required operating parameters, the performance evaluation tests were conducted. After the tests, the jujube fruit, impurities, and the unpicked jujube fruit in the test areas were collected and weighed.

According to the operational requirements and test standards, the pickup rate $Y_1$, impurities rate $Y_2$, and working efficiency $Y_3$ were determined as the evaluation indexes. The calculation methods are shown in Equations (11)–(13):

$$Y_1 = \frac{m_{j.j}}{m_{g.j} + m_{j.j}} \times 100\% \tag{11}$$

$$Y_2 = \frac{m_{i.j}}{m_{j.j} + m_{i.j}} \times 100\% \tag{12}$$

$$Y_3 = \frac{m_{j.j}}{t} \tag{13}$$

where $Y_1$ is the pickup rate, %; $Y_2$ is the impurities rate, %; $Y_1$ is the working efficiency, kg·h$^{-1}$; $m_{g.j}$ is the mass of jujube fruit that was not picked up by the PSH, kg; $m_{j.j}$ is the mass of picked up jujube fruit, kg; $m_{i.j}$ is the mass of the impurities, kg; $t$ is pure working time, h.

The forward speed, provoke teeth buried depth, and airflow velocity were taken as the test factors. According to the pre-tests and the previous analysis, their levels were selected at 0.15–0.35 m·s$^{-1}$, 30–90 mm, and 25–35 m·s$^{-1}$, respectively.

A three-factor and three-level response surface research method was used to evaluate the PSH performance with the Box–Behnken method [32] of Design-Expert software (version 10.0.3, Stat-Ease Inc., Minneapolis, MN, USA), and the test factors and levels are shown in Table 2.

**Table 2.** Arrangements of the test factors and levels with the Box–Behnken method.

| Levels | Factors | | |
| --- | --- | --- | --- |
| | Forward Speed $X_1$/m·s$^{-1}$ | Provoke Teeth Buried Depth $X_2$/mm | Airflow Velocity $X_3$/m·s$^{-1}$ |
| −1 | 0.25 | 30 | 25 |
| 0 | 0.30 | 60 | 30 |
| 1 | 0.35 | 90 | 35 |

The Box–Behnken method was used to code the test factors and levels, and the tests were carried out according to the test schedules. Table 3 shows the test schedules and results, which are a total of 17 groups of tests, including 12 groups of analysis factors and 5 groups of zero estimation errors.

**Table 3.** Experimental schemes and results with the Box–Behnken method.

| No. | Factors | | | Indexes | |
| --- | --- | --- | --- | --- | --- |
| | Forward Speed $X_1$/m·s$^{-1}$ | Provoke Teeth Buried Depth $X_2$/mm | Airflow Velocity $X_3$/m·s$^{-1}$ | Pickup Rate $Y_1$/% | Impurities Rate $Y_2$/% |
| 1 | 0.15 | 30 | 30 | 97.15 | 5.7 |
| 2 | 0.35 | 30 | 30 | 96.34 | 7.4 |
| 3 | 0.15 | 90 | 30 | 99.020 | 8.2 |

**Table 3.** *Cont.*

| No. | Factors | | | Indexes | |
| --- | --- | --- | --- | --- | --- |
| | Forward Speed $X_1$/m·s$^{-1}$ | Provoke Teeth Buried Depth $X_2$/mm | Airflow Velocity $X_3$/m·s$^{-1}$ | Pickup Rate $Y_1$/% | Impurities Rate $Y_2$/% |
| 4 | 0.35 | 90 | 30 | 98.59 | 9.5 |
| 5 | 0.15 | 60 | 25 | 98.76 | 5.6 |
| 6 | 0.35 | 60 | 25 | 97.77 | 6.5 |
| 7 | 0.15 | 60 | 35 | 99.97 | 12.4 |
| 8 | 0.35 | 60 | 35 | 99.92 | 14.8 |
| 9 | 0.25 | 30 | 25 | 95.76 | 4.5 |
| 10 | 0.25 | 90 | 25 | 99.37 | 6.6 |
| 11 | 0.25 | 30 | 35 | 98.52 | 11.1 |
| 12 | 0.25 | 90 | 35 | 99.86 | 15.2 |
| 13 | 0.25 | 60 | 30 | 99.34 | 6.4 |
| 14 | 0.25 | 60 | 30 | 99.41 | 6.5 |
| 15 | 0.25 | 60 | 30 | 98.97 | 6.9 |
| 16 | 0.25 | 60 | 30 | 99.33 | 6.7 |
| 17 | 0.25 | 60 | 30 | 99.26 | 6.8 |

## 3. Results and Discussion

The Analysis module in the Design-Expert 10.0.3 software was used to analyze the variance of the experimental data in Table 3, and the results of the variance analysis of the pickup rate, impurities rate, and working efficiency were shown in Table 4.

**Table 4.** The variance analysis for pickup rate, impurities rate, and working efficiency.

| Source of Variance | Pickup Rate | | | Impurities Rate | | | Working Efficiency | | |
| --- | --- | --- | --- | --- | --- | --- | --- | --- | --- |
| | Sum of Squares | Means Square | *p* Value | Sum of Squares | Means Square | *p* Value | Sum of Squares | Means Square | *p* Value |
| Model | 24.00 | 2.67 | <0.0001 ** | 164.50 | 18.28 | <0.0001 | $4.65 \times 10^6$ | $5.17 \times 10^5$ | <0.0001 ** |
| $X_1$ | 0.65 | 0.65 | <0.0023 ** | 4.96 | 4.96 | 0.0001 ** | $4.60 \times 10^6$ | $4.60 \times 10^6$ | <0.0001 ** |
| $X_2$ | 10.28 | 10.28 | <0.0001 ** | 14.58 | 14.58 | <0.0001 ** | 2876.61 | 2876.61 | 0.3590 |
| $X_3$ | 5.46 | 5.46 | <0.0001 ** | 114.76 | 114.76 | <0.0001 ** | 24.50 | 24.50 | 0.9304 |
| $X_1X_2$ | 0.036 | 0.036 | 0.3095 | 0.040 | 0.040 | 0.4805 | 228.01 | 228.01 | 0.7903 |
| $X_1X_3$ | 0.22 | 0.22 | 0.0302 * | 0.56 | 0.56 | 0.0268 * | 4013.22 | 4013.22 | 0.2843 |
| $X_2X_3$ | 1.29 | 1.29 | 0.0003 ** | 1.00 | 1.00 | 0.0074 ** | 8807.82 | 8807.82 | 0.1296 |
| $X_1{}^2$ | 0.61 | 0.61 | 0.0028 ** | 2.42 | 2.42 | 0.0007 ** | 30762.00 | 30762.00 | 0.0149 * |
| $X_2{}^2$ | 5.16 | 5.16 | 0.0026 ** | 0.34 | 0.34 | 0.0677 | 1163.75 | 1163.75 | 0.5522 |
| $X_3{}^2$ | 0.21 | 0.21 | 0.0336 * | 24.40 | 24.40 | 0.0001 ** | 8621.32 | 8621.32 | 0.1331 |
| Residual | 0.21 | 0.030 | | 0.50 | 0.072 | | 20902.54 | 2986.08 | |
| Lack of fit | 0.093 | 0.031 | 0.4630 | 0.33 | 0.11 | 0.1913 | 15047.36 | 5015.79 | 0.1326 |
| Pure error | 0.12 | 0.029 | | 0.17 | 0.043 | | 5855.18 | 1463.79 | |
| Total | 24.21 | | | 165.00 | | | $4.67 \times 10^6$ | | |

Where ** indicates extremely significant factors ($p < 0.01$); * indicates the significant factors ($0.01 < p \leq 0.05$).

### 3.1. Pickup Rate

The variance analysis for pickup rate indicates that the fitting degree of the regression equation model of $Y_1$ is extremely significant ($p < 0.001$) and the lack of fit is 0.4630, which is not significant (Table 4). Hence, the predicted values are highly correlated with the actual values, and the model can be utilized to analyze and predict the pickup rate.

The quadratic regression fitting analysis was conducted on the results of Table 4 with Design-Expert 10.0.3. The regression equation of pickup rate can be obtained as follows:

$$Y_1 = 91.16 + 0.14X_1 + 0.29X_2 - 0.26X_3 + 0.032X_1X_2 + 0.47X_1X_3 \\ -0.0038X_2X_3 - 37.98X_1{}^2 - 0.0012X_2{}^2 + 0.0089X_3{}^2 \tag{14}$$

To intuitively analyze the influence of factor interaction on pickup rate, the response surface of the pickup rate regression equation was drawn (Figure 7).

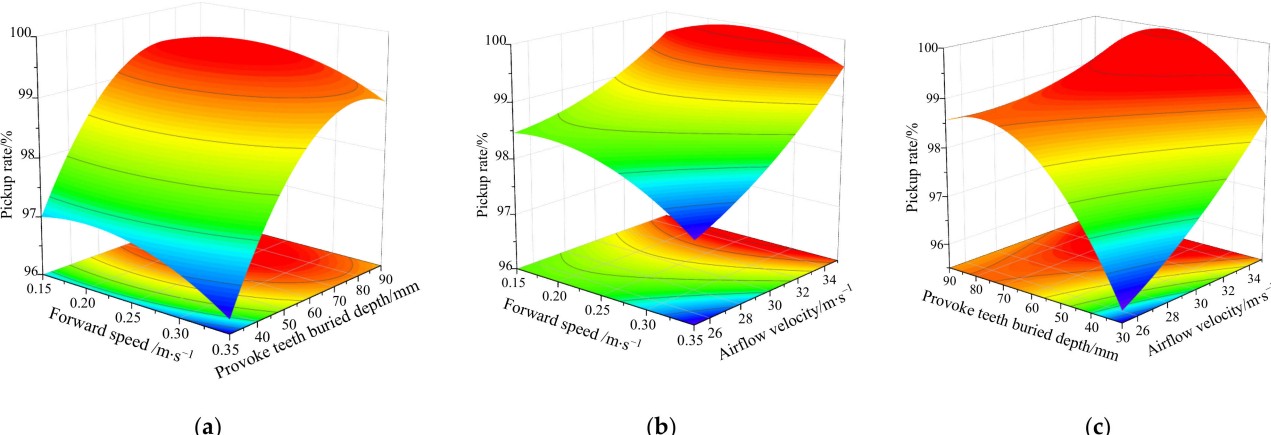

(**a**)  (**b**)  (**c**)

**Figure 7.** Interaction factors effect on the pickup rate: (**a**) interaction factors between the forward speed and the provoke teeth buried depth, (**b**) interaction factors between the forward speed and the airflow velocity, and (**c**) interaction factors between the provoke teeth buried depth and the airflow velocity.

Figure 7a–c are the response surface diagrams of the influence of the forward speed and the provoke teeth buried depth, the forward speed, and the airflow velocity, and the provoke teeth buried depth and the airflow velocity on the pickup rate.

The pickup rate decreases gradually with the increase of forward speed, and the change trend was relatively gentle between 0.15 and 0.25 m·s$^{-1}$, and the trend increases obviously as the forward speed was greater than 0.25 m·s$^{-1}$. The pickup rate first increases rapidly and then tends to be flat with the increase of the provoke teeth buried depth. The pickup rate increases rapidly with the increase of airflow velocity, and the increase trend was further intensified as the airflow velocity was greater than 32 m·s$^{-1}$.

The reason may be that the time of airflow acting on the jujube fruit was reduced, with the increase of forward speed, resulting in the reduction of pickup rate. However, the effect was not significant. More jujube fruit in pits will pick up with the increase of the provoke teeth buried depth, so the probability of unpicked jujube fruit will be reduced. The higher airflow velocity has a greater force on jujube fruit and a large action range, which can suck up more jujube fruit. Therefore, the airflow velocity has a significant impact on the pickup rate.

### 3.2. Impurities Rate

The variance analysis for impurities rate indicates that the fitting degree of regression equation model of $Y_2$ is extremely significant ($p < 0.001$) and the lack of fit is 0.1913, which is not significant (Table 4). Hence, the predicted values are highly correlated with the actual values, and the model can be used to analyze and predict the impurities rate.

The regression equation of impurities rate can be obtained:

$$Y_2 = 82.93 - 50.50X_1 - 84.33X_2 - 5.41X_3 - 33.33X_1X_2 + 0.75X_1X_3 \\ + 3.33X_2X_3 + 75.75X_1^2 + 313.89X_2^2 + 0.096X_3^2 \tag{15}$$

Figure 8 shows the response surface of the impurities rate regression equation. Figure 8a–c are the response surface diagrams of the influence of the forward speed and the provoke teeth buried depth, the forward speed, and the airflow velocity, and the provoke teeth buried depth and the airflow velocity on the impurities rate.

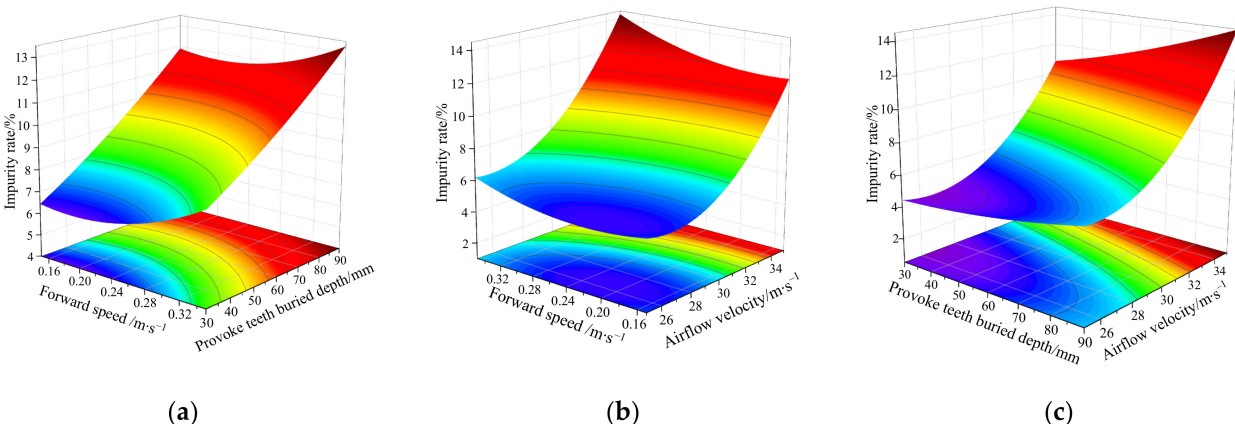

**Figure 8.** Interaction factors' effect on the impurities rate: (**a**) interaction factors between the forward speed and the provoke teeth buried depth, (**b**) interaction factors between the forward speed and the airflow velocity, and (**c**) interaction factors between the provoke teeth buried depth and the airflow velocity.

The impurities rate first decreases slowly and then increases rapidly with the increase of forward speed. The impurities rate was at its minimum value when the forward speed was about 0.20 m·s$^{-1}$ and the provoking teeth buried depth was at its minimum value. The impurities rate increases linearly with the increase of the provoke teeth buried depth. The impurities rate increases gradually with the increase of airflow velocity, when the airflow velocity was less than 30 m·s$^{-1}$, and the trend increases rapidly when the airflow velocity was greater than 30 m·s$^{-1}$.

The reason may be that with the increase of the forward speed, the airflow acting time on jujube fruit was less, so the absorbed impurities were relatively reduced when the forward speed was lower than 0.20 m·s$^{-1}$. The provoking teeth will break the massive soil and then be sucked up, increasing the impurities rate, when the forward speed was greater than 0.20 m·s$^{-1}$. The deeper the provoking teeth buried depth, the more soil is disturbed by the provoking teeth, and then it will be sucked up together with the jujube fruit, so the impurities rate increases linearly with the provoking teeth buried depth. The reason the airflow velocity has the most significant effect on the impurities rate is that the greater airflow velocity will inhale more soil blocks. The aerodynamic equivalent diameter of the soil blocks was similar to the jujube fruit, which were difficult to remove.

### 3.3. Working Efficiency

The variance analysis for working efficiency indicates that the fitting degree of the regression equation model of $Y_3$ is extremely significant ($p < 0.001$) and the lack of fit is 0.1326, which is not significant (Table 4). Hence, the predicted values are highly correlated with the actual values, and the model can be utilized to analyze and predict the working efficiency.

The regression equation of working efficiency can be obtained:

$$Y_3 = 2323.01 + 10106.13X_1 - 11.60X_2 - 143.56X_3 - 2.52X_1X_2 + 63.35X_1X_3 \\ + 0.31X_2X_3 - 8547.50X_1{}^2 + 0.018X_2{}^2 + 1.81X_3{}^2 \tag{16}$$

Figure 9 indicates the response surface of the working efficiency regression equation. Figure 9a–c are the response surface diagrams of the influence of the forward speed and the provoke teeth buried depth, the forward speed, and the airflow velocity, and the provoke teeth buried depth and the airflow velocity on the working efficiency.

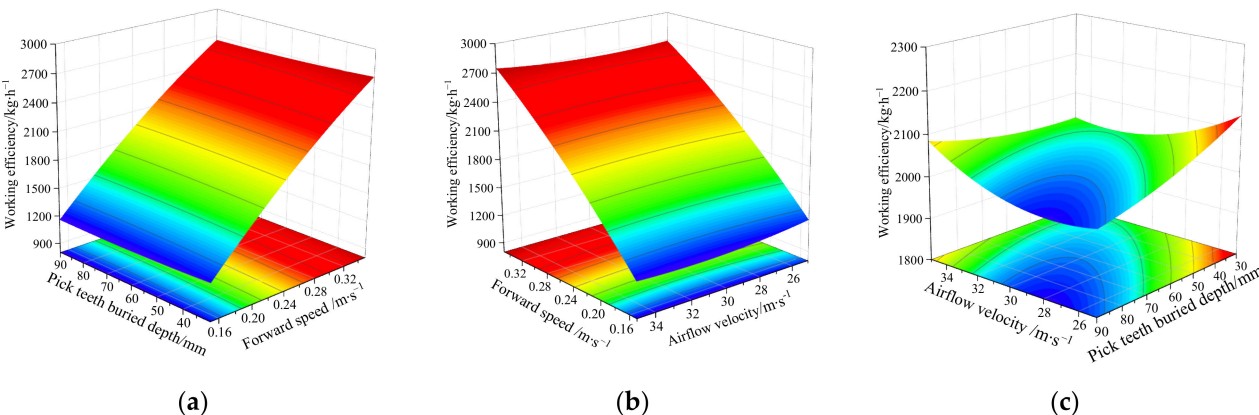

(a)                 (b)                 (c)

**Figure 9.** Interaction factors' effect on the working efficiency: (**a**) interaction factors between the forward speed and the provoke teeth buried depth, (**b**) interaction factors between the forward speed and the airflow velocity, and (**c**) interaction factors between the provoke teeth buried depth and the airflow velocity.

The working efficiency showed an approximately linear trend from 600 kg·h$^{-1}$ to 1300 kg·h$^{-1}$, as the forward speed increases. With the change of airflow velocity and provoke teeth buried depth, the variation range of the working efficiency only changes from 2000 kg·h$^{-1}$ to 2150 kg·h$^{-1}$. The reason may be that the jujube fruit in the area to be collected were almost evenly distributed, so the working efficiency was only related to the forward speed.

### 3.4. Parameter Optimization

The full factor quadratic regression model of indicators with the optimization module of Design-Expert 10.0.3 software was used to obtain the optimal combination of experimental factors [33–35]. The constraint condition and the experimental factors indexes were as follows:

$$\begin{cases} \max Y_1 = (X_1, X_2, X_3) \\ \min Y_2 = (X_1, X_2, X_3) \\ \max Y_3 = (X_1, X_2, X_3) \\ s.t. \begin{cases} X_1 \in (0.15, \ 0.35) \\ X_2 \in (30, \ 90) \\ X_3 \in (25, \ 35) \end{cases} \end{cases} \tag{17}$$

The optimal parameter combination was obtained that the forward speed, provoke teeth buried depth, and airflow velocity were 0.21 m·s$^{-1}$, 74 mm, and 26.4 m·s$^{-1}$, respectively. Additionally, the pickup rate, impurities rate, and working efficiency were 99.36%, 5.63%, and 1672.3 kg·h$^{-1}$, respectively.

To verify the parameter optimization results and evaluate the PSH performance, the field verification tests were carried out based on the above test conditions in the whole process mechanization demonstration base of jujube orchard in group 13, Alar city, Xinjiang Province, China. The field verification test process and work performance are shown in Figure 10.

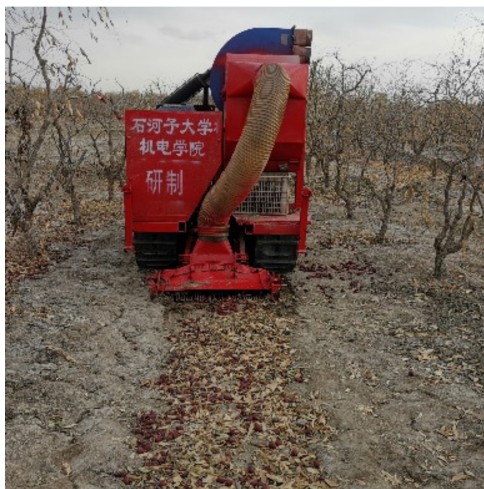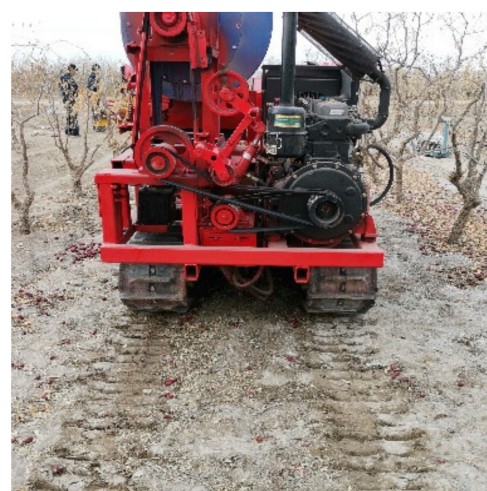

**Figure 10.** Field verification test process and work performance of the provoke-suction type harvester for ground jujube fruit (PSH). The Chinese in left image means "developed by the College of Mechanical and Electrical Engineering, Shihezi University".

The field verification tests were repeated five times with the optimal parameter combination, and the arithmetic average values were taken as the results. The verification test results are shown in Table 5.

**Table 5.** Comparison of the optimal results and the verification test results for evaluation indexes.

| Evaluation Indexes | Optimal Results | Verification Test Results | Relative Errors |
|---|---|---|---|
| Pickup rate | 99.36% | 98.05% | 1.32% |
| Impurities rate | 5.63% | 5.97% | 6.04% |
| Working efficiency | 1672.3 kg·h$^{-1}$ | 1591.2 kg·h$^{-1}$ | 4.85% |

The field verification test results showed that the pickup rate, impurities rate, and working efficiency of the PSH were 98.05%, 5.97%, and 1672.3 kg·h$^{-1}$, respectively. Furthermore, the relative errors with the parameter optimization values were 1.32%, 6.04%, and 4.85%, respectively, indicating that the parameter optimization model can accurately predict the test results.

*3.5. Discussion*

The pickup rate of jujube fruit harvester developed by Lu [11] is 93.6%, and Zhang et al. [13] is 96.41%. The pickup rates of Jun jujube fruit and Grey jujube fruit are 92.20% and 90.65% by Zhang [15]. In this study, the pickup rate is 98.05%, which has significantly improved. The reason was that the provoking teeth can shovel the jujube fruit in low-lying areas that were difficult to pick up by the negative pressure airflow action.

The impurities rate of jujube fruit harvester developed by Zhang et al. [13] is 1.54%, and the impurities rate Jun jujube fruit and Grey jujube fruit are 3.75% and 4.28%, which studied by Zhang [15]. The impurity rate was 5.97% in our study, higher than the existing studies. The reason may be that the provoke teeth not only shovel the jujube fruit but also shovel some soil blocks. These soil blocks were difficult to remove since the aerodynamic equivalent diameter of the soil blocks was similar to the jujube fruit. However, there was a large size difference between soil blocks and jujube fruit, which were easy to remove by screening in subsequent operations.

The existing studies reported that the working efficiency was in the range of 100–500 kg·h$^{-1}$ [16–20]. In our paper, the working efficiency was 1591.2 kg·h$^{-1}$, which was improved more than three times. One reason was that after the provoke teeth gather the jujube fruit, the negative pressure airflow can act on the jujube fruit more intensively, reducing the dissipation of airflow. The other reason was that the friction drag force between

the airflow and the jujube fruit was the main force [28], as the jujube fruit were suctioned by the negative pressure airflow. In this study, the pressure drag was the main force when the jujube fruit were suctioned along the provoke teeth. The pressure drag has more force on the jujube fruit than the friction drag at a higher Reynolds number.

## 4. Conclusions

In this paper, a provoke-suction type harvester for ground jujube fruit (PSH) was designed, based on the principle of suctioning jujube fruit after shoveling. The Box–Behnken method was used to evaluate the performance of the PSH. The results showed that the pickup rate, impurity rate, and working efficiency were 99.36%, 5.63%, and 1672.3 kg·h$^{-1}$, respectively, as the forward speed, provoke teeth buried depth and airflow velocity were 0.21 m·s$^{-1}$, 74 mm, and 26.4 m·s$^{-1}$, respectively. Furthermore, the field verification tests were carried out according to the optimal parameter combination conditions, of which results showed that the pickup rate, impurity rate, and working efficiency were 98.05%, 5.97%, and 1591.2 kg·h$^{-1}$, respectively. The operation performance of the PSH meets the requirements of jujube fruit harvesting, and the picking efficiency was significantly improved compared with the traditional aspirated-air type jujube fruit pickup machine. This research can provide a new mechanized operation method for picking up jujube fruit and a reference for the development of a jujube fruit harvester.

In the future, this research can be further improved in two aspects. First, the structure of the cleaning device should continue to be improved, to reduce the impurity rate. Second, future studies should focus on the damage rate of the PSH.

**Author Contributions:** Conceptualization, methodology, data curation, formal analysis, writing—original draft, writing—review and editing, G.S. and J.L.; investigation, H.D.; data curation, L.D.; funding acquisition, J.L., L.D. and L.W.; validation, L.Z.; supervision, Z.K. All authors have read and agreed to the published version of the manuscript.

**Funding:** This research was funded by the National Natural Science Foundation of China (52165037; 51865050), and the Regional Innovation Guidance Plan of the Xinjiang Production and Construction Corps (2021BB003).

**Institutional Review Board Statement:** Not applicable.

**Informed Consent Statement:** Not applicable.

**Data Availability Statement:** All data are presented in this article in the form of figures and tables.

**Conflicts of Interest:** The authors declare no conflict of interest.

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
