# Peer review of "Design and Parameters Optimization of a Provoke-Suction Type Harvester for Ground Jujube Fruit"

_agriculture, doi:10.3390/agriculture12030409_

Round 1
Reviewer 1 Report
This is a very good manuscript. Below are some comments for improvement.
- The title is clear and to the point
- Abstract, Line 13: Twisted statement, please re-word.
- The objective statement is clear and valid. The abstract is very well organized and written.
- Line 31: Error reference. Please double-check for similar flaws.
- Introduction, Opening paragraph: You need to provide 2-3 lines of background about robotic harvesting. Please consider adding the following motivational statements and background: "Traditional harvesting of fruits is a labor-intensive......Modering agriculture is shifting from tedious manual harvesting to a continuously automated operation..[REF1,REF2]"
Ref 1: Shamshiri, R. R., Hameed, I. A., Karkee, M., & Weltzien, C. (2018). Robotic harvesting of fruiting vegetables: A simulation approach in V-REP, ROS and MATLAB. Proceedings in Automation in Agriculture-Securing Food Supplies for Future Generations, 126.
Ref 2: Kootstra, G., Wang, X., Blok, P. M., Hemming, J., & Van Henten, E. (2021). Selective harvesting robotics: current research, trends, and future directions. Current Robotics Reports, 2(1), 95-104.
Figure 1: Please provide a general caption describing the figure before mentioning the labels.
Line 105, 175, 183, 183,247 ...: where, please double check for similar flaws.
Figure 2: Can you provide a color version of this figure?
Line 186: Did you mean "Design of the cleaning device"?
Table 4, Table 5, Table 6 I don't think you need to provide such detailed results of statistical analysis. It is not of interest for many readers the degree-of-freedom, and F-value. Please consider removing these tables...or at least provide a summary of all three tables as one Table. Again you don't need to report F-vaues, etc. You can simply highlight the core findings in your text.
Reviewer 2 Report
The present paper introduces an interesting technique for harvesting Jujube fruit through Provoke-suction.
The machine is well detailed, and the papers present a good description of the harvesting method, achieving an excellent working efficiency.
The authors must improve the general English of the paper. I suggest the usage of orthography software (e.g. Grammarly, InstaText etc).
I would like to see a wider literature review. There are only a few references in the introduction. There is not even a related work section. So please, improve the literature review.
